# Effects of the Feldenkrais Method as a Physiotherapy Tool: A Systematic Review and Meta-Analysis of Randomized Controlled Trials

**DOI:** 10.3390/ijerph192113734

**Published:** 2022-10-22

**Authors:** Rémi Berland, Elena Marques-Sule, José Luis Marín-Mateo, Noemi Moreno-Segura, Ana López-Ridaura, Trinidad Sentandreu-Mañó

**Affiliations:** 1Department of Physiotherapy, University of Valencia, 46010 Valencia, Spain; 2Physiotherapy in Motion, Multispeciality Research Group (PTinMOTION), University of Valencia, 46010 Valencia, Spain

**Keywords:** feldenkrais, physical therapy, elderly, musculoskeletal pain, multiple sclerosis, parkinson’s disease

## Abstract

The Feldenkrais Method (FM) is based on the learning of alternative movement patterns, carried out in an active and conscious way, which may have therapeutic effects. The objective of this systematic review is to identify the populations and conditions for which the FM can be used in physiotherapy and to determine the intervention modalities. Research in PubMed, Cochrane and PEDro databases was performed. The PEDro scale was employed to assess the methodological quality. Meta-analyses (MA) were performed whenever populations and outcome measures were comparable in at least two studies. Sixteen studies were included. In elderly people, in three of the four selected trials, the FM group significantly improved gait, balance, mobility and quality of life. The MA showed significant differences between interventions in the Timed-Up-and-Go test [Cohen’s d = −1.14, 95% CI (−1.78, −0.49), *p* = 0.0006]. FM significantly improved pain, functional balance, and perceived exertion in three trials performed on subjects with cervical, dorsal, or shoulder pain. FM demonstrated improvements in pain, disability, quality of life and interoceptive awareness in the three trials performed in subjects with chronic low back pain. In multiple sclerosis, an improvement in functional capacity was observed in the two selected studies. The MA showed no significant differences between groups in the Function (*p* = 0.97) and Control (*p* = 0.82) dimensions of the Multiple Sclerosis Self-Efficacy Scale. In Parkinson’s disease, two studies showed significant effects on quality of life and functional tests. In conclusion, evidence shows that FM has therapeutic effects comparable to other physiotherapy techniques in patients with spine pain. In addition, improvements in mobility and balance were seen in the elderly and people with neurodegenerative diseases.

## 1. Introduction

The Feldenkrais Method (FM) is a technique aiming at increasing personal self-knowledge through conscious movements, developed by the physicist Moshe Feldenkrais [1,2]. This method is based on the discovery and learning of varied, alternative patterns of movement and aims to improve the human ability to learn movement [3,4]. The method has been applied to different educational areas in different countries to improve sports and theatre performance [5]. Furthermore, FM has been employed in movement lessons by using two modalities. One modality, named “Awareness through movement” (ATM), is applied in group sessions. In this modality, the Feldenkrais therapist guides the subjects verbally into postures and movements and asks them about bodily sensations. The other modality, named “Functional Integration” (FI), consists of individual sessions of passive and soft mobilizations directed to movement re-education and to improve proprioception [5,6]. 

The main objective of FM seems to be more educational than therapeutic [7]. However, the FM is recognized by the German health system and is widely employed in the United States, Australia and Germany. In Germany, FM is generally used to treat postural deformities, musculoskeletal limitations, dorsal pain, neurological pathologies, chronic pain, development disorders in children and adolescents, psychosomatic and stress-related disorders [1]. Thus, the indications of the FM seem to be compatible with the physical therapy practice. 

Previous systematic reviews have studied the effects of FM. The first of them was published in 2005 by Ernst and Canter [2], and it concluded that although there were some trials that supported the FM, the evidence published was not sufficient in quantity and quality to support the method robustly. The second systematic review was published in 2015 by Hillier and Worley [5], where the authors updated the first review and included a meta-analysis (MA), concluding that the trials about FM were very heterogeneous in terms of population, variables and results. They suggested that although there was promising evidence of the efficacy of the FM in improving balance in elderly people, more research was needed to support these findings. Another systematic review focused on patients with cervical or lumbar pain [8] included four trials and concluded that there was some evidence about the efficacy of FM to improve pain, although they determined that the intervention modalities seemed to be inconsistent and the measurement variables heterogeneous. 

Accordingly, the hypothesis of this systematic review and MA is that the actual evidence would be sufficient to support the inclusion of FM among physical therapy tools, and the FM could have similar effects when compared to other techniques already employed in physical therapy. 

To test this hypothesis, it is necessary to update the evidence about FM, focusing on its therapeutic potential within the scope of physical therapy. Therefore, the objectives of this systematic review and MA were (1) to identify the recent scientific evidence about FM and determine the populations and pathologies where FM could be employed as a physical therapy tool; (2) to analyze which of the two modalities of FM is more effective in physical therapy; (3) and to analyze the therapeutic effects obtained with the FM. 

## 2. Materials and Methods

This review was planned and conducted based on the Preferred Reporting Items for Systematic Review and Meta-Analyses (PRISMA) [9]. The protocol of this review has been previously registered in the International Prospective Register of Systematic Reviews (PROSPERO) (CRD42021282249). 

### 2.1. Eligibility Criteria

This systematic review included all the randomized clinical trials (RCT) that met the following eligibility criteria: (1) RCT with a population of interest in the field of physical therapy; (2) group, individual or mixed-modality sessions of FM as the primary intervention; (3) a control group (CG) classified as no intervention, placebo, simulation, educational intervention or conventional physical therapy intervention; (4) at least one primary outcome measure to assess patient’s physical condition such as mobility, cardiovascular condition, muscle strength, balance, pain, gait performance, functionality or quality of life; (5) articles written in English or in Spanish.

As an exclusion criterion, studies carried out on healthy subjects were excluded from this review.

### 2.2. Search Strategy and Selection Process

The following electronic databases were searched: PubMed, Cochrane and PEDro, from inception to August 2022. Reference lists of selected trials, previous publications and cited reviews were manually examined.

The following keywords were employed to carry out the search: “Feldenkrais”, “Awareness through Movement” and “Functional Integration”. Terms were related using the Boolean index “OR”. No publication data restrictions were applied. Results were restricted only to human subjects. 

The searching process, the text screening and the application of the eligibility criteria were performed by two independent reviewers (RB, TSM). First, the results were screened by title and abstract, and second, remaining references were assessed for eligibility by in-depth reading of the full texts. Inconsistences between reviewers were solved by consensus. 

### 2.3. Data Collection and Risk of Bias Assessment

Two independent reviewers (RB, TSM) performed the study identification, data extraction and assessed risk of bias using PEDro scale. When different results were obtained by the reviewers, it was discussed among them, and an agreement was reached. 

All the included articles underwent a protocolized, systematic and standardized analysis. The following data were extracted into a spreadsheet: title, authors, year of publication, country, study design, population, sample size, sample mean age, sample gender, characteristics of FM and CG interventions, follow-up, outcome measures and results. 

Regarding PEDro scale, it is considered that scores ≤3 points indicate poor methodological quality, between 4 and 5 points indicate fair methodological quality, between 6 and 8 points mean good methodological quality and scores >8 points indicate excellent methodological quality [10]. 

### 2.4. Statistical Analysis

When RCTs investigated the same type of population and extracted the same outcome measure, a random-effects MA was performed, with standardized mean differences for continuous outcomes. The Review Manager software (RevMan) version 5.4, The Cochrane Collaboration, London, United Kingdom, was employed [11]. 

Moreover, 95% confidence intervals (CI) and heterogeneity values (I^2^) were calculated. I² values of 25%, 50% and 75% were considered as low, moderate and high heterogeneity, respectively [12]. In addition, mean effects sizes and the resulting forest plots to visualize MA results were obtained.

## 3. Results

### 3.1. Study Selection

A total of 330 publications were identified in databases using the search strategy (Figure 1), among which 270 were found in PubMed, 42 in Cochrane and 18 in PEDro. A further 24 trials were found by manual search. After applying the eligibility criteria and discarding duplicates, 16 studies were finally included. 

### 3.2. Methodological Quality Assessment

The methodological quality of the included studies is presented in Table 1. Five articles [13,14,15,16,17] presented a good methodological quality, one of them [18] presented a poor methodological quality, and the others presented a fair methodological quality. Neither the subjects nor the therapists were blinded in any of the studies. Intention-to-treat analysis was performed in four studies [13,14,19,20] and allocation concealment in five studies [13,15,16,17,21]. In most studies, assessors were not blinded, nor was the follow-up adequate.

### 3.3. Characteristics of the Studies

Table 2 shows the characteristics and results of the included studies. The articles were published between 1994 [19] and 2020 [13]. Most studies were RCTs with two parallel groups. The populations found were heterogeneous. Three of the studies investigated subjects with cervical, dorsal or shoulder pain [14,19,23], three of them subjects with chronic low back pain [13,15,16], two of them subjects with multiple sclerosis [22,26], two of them subjects with Parkinson’s disease [27,28], four of them elderly subjects [17,18,20,24], one trial investigated children with nocturnal bruxism [25] and one investigated middle-aged subjects with intellectual disability [21].

The most frequent intervention in the Feldenkrais group (FG) was ATM. Two trials employed a combination of ATM and FI [14,23]. The CG performed no intervention in six trials [14,17,18,20,21,23], educational sessions in four trials [13,26,27,27], simulation in two trials [19,22], physical activity in two trials [13,24] and conventional physical therapy in one trial [15]. In the three-arms clinical trials, the other intervention group performed physical therapy sessions [23] or Pilates [24].

Regarding outcome measures, the most frequently employed were pain, balance, functional capacity and quality of life. 

The characteristics of the studies for each type of population investigated are detailed below.

#### 3.3.1. Elderly Subjects 

The effects of FM in elderly subjects were studied in four trials (total number of participants n = 286) [17,18,20,24]. The mean age was between 69 and 76 years, and the predominant gender was female. The intervention protocol of the FG was one to three ATM group sessions per week, totaling between eight sessions [17] and eighteen sessions [24]. The CG performed either no intervention or a 12-min walking program at a comfortable speed with a previous warm-up [24]. Nambi et al. [24] included a second intervention group that performed 18 sessions of Pilates.

Regarding outcome measures, all trials evaluated mobility and balance. Except for Ullmann et al. [20], all trials evaluated the quality of life. 

Regarding trial results, three of the four trials [17,20,24] obtained significant improvements in favor of FM with respect to mobility (Timed-Up-and-Go test, gait speed), balance (tandem posture, functional reach and fear of falling) [17,18,20,24] and quality of life [18,24]. The Timed-Up-and-Go test, analyzed in all studies, showed significant improvements in all trials except for Palmer [18]. The MA showed significant differences between interventions (FG and CG) in The Timed-Up-and-Go test [Cohen’s d = −1.14, 95% CI (−1.78, −0.49), *p* = 0.0006]. Complete MA information with forest plot is presented in Figure 2. 

Heterogeneity (I^2^) was low, with a value of 5%. As far as quality of life is concerned, significant improvements were found in favor of the FM intervention, except in the study of Vrantsidis et al. [17]. It was not possible to perform a MA because of the heterogeneity of employed outcome measures. 

#### 3.3.2. Subjects with Cervical, Dorsal or Shoulder Pain

Three trials investigated the effects of FM in patients with cervical, dorsal or shoulder pain (total number of participants n = 181) [14,19,23]. The mean age was between 18 and 59 years, and most of the participants were female. The FG intervention protocol consisted of: (1) a program of 12–16 sessions of ATM combined with FI [14,23] or (2) a single pre-recorded ATM session [19]. Moreover, Lundblad et al. [23] included a second intervention group that performed a physical therapy protocol for 16 weeks. The CG performed either no intervention [14,23] or a pre-recorded simulated session of ATM [19].

The outcome measures related to pain [14,23], functional balance [19] and perceived exertion [19] were heterogeneous; thus, the MA was not performed. Significant changes were found in favor of the FG [14,19,23]. Regarding pain and muscle complaints, significant differences were found between FG and CG [14,23]. Lundqvist et al. [14] observed a significant increase in palpation pain in the occipital muscles and left and right upper trapezius in the CG at post-intervention and at one-year follow-up, while the FG did not show significant changes. Lundqvist et al. [14] showed a significant reduction of muscle complaints in the FG. Furthermore, Lundblad et al. [23] showed a significant reduction of muscle complaints and disability in FG, while the muscle complaints increased in the CG and remained stable in the physical therapy group. Regarding pain intensity, Lundblad et al. [23] showed a significant reduction in CG and FG, but the reduction was more pronounced in FG compared to CG. The physical therapy group did not show significant reductions in pain intensity. Lastly, Chinn et al. [19] showed a significant reduction in the perceived exertion during the forward reach test in the FG. 

#### 3.3.3. Subjects with Chronic Low Back Pain 

Three trials investigated the effects of FM in subjects with chronic low back pain (total number of participants n = 139) [13,15,16]. The mean age was between 39 and 61 years, and most of the sample was female. The FG performed either 10 ATM group sessions [13,15], or a unique, pre-recorded ATM session [16]. Regarding the CG, Paolucci et al. [15] provided back school lessons imparted by a physical therapist and Ahmadi et al. [13] educational sessions and core stability exercises. 

Regarding outcome measures, Paolucci et al. [15] and Ahmadi et al. [13] assessed pain, disability, quality of life and interoceptive awareness. Both trials evaluated pain through the McGill questionnaire and interoceptive awareness through the Multidimensional Assessment of Interoceptive Awareness questionnaire [29]. Smith et al. [16] assessed pain and anxiety. All the studies showed significant changes in the FG [13,15,16]. Regarding pain, changes were found in the FG, but no differences with the CG were found. The FG obtained better scores than the back school lessons with respect to the McGill Present Pain Intensity subscale [15] and obtained better McGill scores than the core stability training program [13]. Smith et al. [16] measured pain through the Short Form- McGill questionnaire and observed a significant reduction in pain in the CG. Moreover, the FM showed significant improvements in quality of life, disability and interoceptive awareness compared to core stability training [13]. On the other hand, Paolucci et al. [15] showed better scores in the back school group than the FG with respect to the Vitality and Social Functioning dimensions of the Short Form-36 (measuring quality of life), while the Multidimensional Assessment of Interoceptive Awareness questionnaire score was not different between groups.

#### 3.3.4. Subjects with Multiple Sclerosis 

Two studies investigated the effects of FM in subjects with multiple sclerosis (total number of participants n = 32) [22,26]. The mean age was between 45 and 56 years, and the predominant gender was female. The intervention protocol consisted of ATM group sessions. Johnson et al. [22] performed a total of 6 h of ATM, while Stephens et al. [26] imparted a total of 20 h. The CG received a manual treatment simulation [22] or educative sessions [26]. Stephens et al. [26] assessed balance and functional capacity, while Johnson et al. [22] evaluated functional capacity, anxiety, depression and symptoms. Both trials evaluated confidence in functional capacity through the Multiple Sclerosis Self-Efficacy scale. 

Regarding the results, both trials showed significant changes in the FG [22,26]. Johnson et al. [22] showed a significant reduction in the perceived stress in the FG. Stephens et al. [26] showed a significant improvement in balance and balance confidence in the FG. Regarding confidence in functional capacity, both trials showed a trend toward improvement in the FG. 

The MA showed no significant differences between FG and CG with respect to the dimensions of Function [Cohen’s d = 4.06, 95% CI (−9.86, 17.98), *p* = 0.97] and Control [Cohen’s d = 9.50, 95% CI (−12.56, 31.56), *p* = 0.82] of the Multiple Sclerosis Self-Efficacy scale. Complete MA information with forest plot is presented in Figure 3A,B.

#### 3.3.5. Subjects with Parkinson’s Disease

The effects of FM in subjects with Parkinson’s disease were analyzed in two studies with identical samples and protocol (total number of participants n = 30) [27,28]. The mean age was 61 years. The FG received a total of 50 h of ATM sessions, while the CG received educative sessions. 

Regarding outcome measures, one of them evaluated the quality of life, depression and cognitive status [27], while the other analyzed balance, mobility, gait speed and strength through functional tests [28]. Both studies showed significant effects on the FG [27,28]. One of them showed significant improvements in quality of life, depression and cognitive status in the FG compared to the CG [27]. The other trial showed that the FG improved significantly in all the variables compared to the CG, whereas in the CG all the values worsened [28]. The Timed-Up-and-Go test showed a reduction in the FG, decreasing from 22.27 ± 2.93 to 12.46 ± 0.92 s (*p* = 0.003).

#### 3.3.6. Other Populations

Quintero et al. [25] investigated the effects of a 30 h ATM sessions protocol on the head posture of 26 children with nocturnal bruxism with a mean age of three to four years. Torres-Unda et al. [21] investigated the effects of a 30 h ATM sessions protocol on the physical function and balance (measured with a stability platform) of middle-aged persons with intellectual disabilities, for whom difficulties due to aging appear earlier than in people without intellectual disabilities [21]. 

Regarding the effects, Quintero et al. [25] showed the efficacy of FM in treating children with nocturn bruxism, for whom the FM increased the craniovertebral angle and corrected the head posture. In middle-aged persons with intellectual disabilities, Torres-Unda et al. [21] showed a significant improvement in physical function in FG compared to CG and a reduction of the sway area.

## 4. Discussion

This systematic review presents the evidence available to date on the effects of FM in the population eligible for physical therapy treatment. Population groups included elderly people, people with musculoskeletal pain in the spine or shoulder, and people with neurodegenerative diseases such as MS or PD.

In relation to the obtained results, most of the trials showed significant changes in the FG. In elderly subjects, the mean differences obtained in favor of FM in the Timed-Up-and-Go test MA (mean difference of 1.14 s) could be considered clinically significant based on the study of Wright et al. [30] where the minimum clinically significant difference in Timed-Up-and-Go test was calculated in patients with hip osteoarthritis and showed that a reductions ≥0.8–1.4 s could be considered clinically significant. Moreover, a reduction in the Timed-Up-and-Go test is related to the decrease in fall risk [31]. This outcome measure is also related to sarcopenia [32], frailty [33] and osteoporosis [34], which are highly prevalent in the elderly population. Future studies should assess whether this treatment method influences this situation of great interest in the elderly population. It shall be noted that the Timed-Up-and-Go test MA results obtained in this review are identical to those obtained by an earlier review [5]. Older adults also improved through the FM in terms of balance, fear of falling and quality of life. Nambi et al. [24] showed that both studied interventions (FM and Pilates method) led to improvements, but the results with the Pilates method were better than with the FM. The authors discuss that the popularity of Pilates and social interactions could partly explain the results obtained. 

Regarding musculoskeletal pain, the subjects with cervical, dorsal or shoulder pain improved through the FM in terms of pain intensity, muscle complaints, leisure disability and perceived exertion during the reach test. Lundblad et al. [23] trial showed that FM was better than conventional therapy. These findings are in line with the results of the Mohan et al. [8] systematic review regarding subjects with chronic low back pain, where FM was equal to or better than physical therapy interventions based on back school [15] or core stability exercises [13]. Significant improvements were found in pain, quality of life, disability, interoceptive awareness and abdominal musculature. According to Mehling et al. [35], interoceptive awareness may play important roles in health, particularly in pain perception. The Multidimensional Assessment of Interoceptive Awareness questionnaire could be useful when the therapeutic intervention directly targets the mind-body interface, as is the case with the FM [35]. Paolucci et al. [15] observed that the effects of FM in reducing pain and improving interoceptive awareness were still effective three months after the start of the intervention. Moreover, they indicated that the FM could reduce the intensity of pain and modify the pain perception faster than back school lessons. Ahmadi et al. [13] concluded that FM has the advantage, compared to an educational program and core exercises, of improving abdominal musculature, interoceptive awareness and disability. 

In subjects with multiple sclerosis, the FM improved the perceived stress, balance and balance confidence. However, the samples were small, and future studies with larger samples are needed to confirm these results. In subjects with Parkinson’s disease, improvements in balance, mobility, speed and quality of life were found. The changes in the Timed-Up-and-Go test (the time spent in the test decreased by 9.8 s to reach a mean value of 12.46 s) indicate that the FM has clinical effects in subjects with Parkinson’s disease. According to Barry et al. [31], values below 13.5 s may indicate a decreased risk of falling. However, since only two studies were published in this area and the sample was the same, new studies are needed to confirm these improvements. 

With regard to methodological quality, which has been criticized in previous reviews [2,5,8], it should be noted that, in the included articles, it was either good or fair, except in one article where the methodological quality was poor. The difficulty of blinding subjects and therapists is common in rehabilitation trials [36]. The trials scores are consistent with the analysis of a study about the construction validity of the PEDro scale, in which Albanese et al. [37] showed that subjects and therapists blinding was almost never implemented, while criteria related to random assignment, comparisons between groups, point measures and variability measures were almost always met. However, only five articles obtained a good methodological quality, and other criteria of the PEDro scale were frequently not met, such as concealed allocation, blinding assessors, adequate follow-up and intention-to-treat analysis. 

In relation to the intervention modality, the most employed protocol for the FG included 10 to 30 sessions, applied once or twice per week, with a duration of 45 min to 2 h. The findings of this review contradict the idea that sessions should be short (between 30–45 min) to prevent mental fatigue [38]. ATM group sessions constituted the most used practice. The improvements obtained through this modality are interesting, considering aspects such as costs and time of the professional’s work. Future cost-effectiveness studies are needed to clarify this issue. Furthermore, the ATM modality enhances participation, which is one of the pillars to consider in a rehabilitation process within the framework of the International Classification of Functioning, Disability and Health [39]. 

The greatest strength of this review is that this is the first work that analyzes the effects of FM on the population that needs physical therapy. In addition, despite the different types of FM and the different control group interventions, the most frequently used exercises and protocols have been described and shortlisted. We also highlight the statistical rigor performed in the study. Moreover, the results of the analyzed variables show a low heterogeneity. Finally, the differentiation by type of illness facilitates the understanding of the results and could be used as a guide for the development of new studies in each field.

This systematic review and meta-analysis have some limitations that should be mentioned. Firstly, the search was performed in a limited number of databases. Secondly, language was also a restriction that could have reduced the number of studies reviewed. Thirdly, the risk of bias remains high due to the insufficient methodological quality of certain studies. Finally, in some populations, trials with larger samples are needed to be able to extract solid conclusions.

## 5. Conclusions

This systematic review and MA concluded that FM applied in ATM group sessions is effective in the treatment and prevention of some pathologies or clinical conditions. Regarding elderly people, FM improves mobility, balance and quality of life. Regarding pain, in people with chronic low back pain, FM has similar benefits such as back school lessons or core stability exercises; in people with cervical pain, the FM may be more adequate than conventional physical therapy. Regarding people with neurodegenerative diseases, the FM is effective in improving balance. 

These findings should be taken with caution due to the low number of RCTs in the different specific populations. Future research would be useful to give more solidity to these findings and to consider the role of this treatment method in the prevention of falls and other age-related problems, such as sarcopenia, frailty and falls, due to its implication in improving the Timed-Up-and-Go test in different populations.

## Figures and Tables

**Figure 1 ijerph-19-13734-f001:**
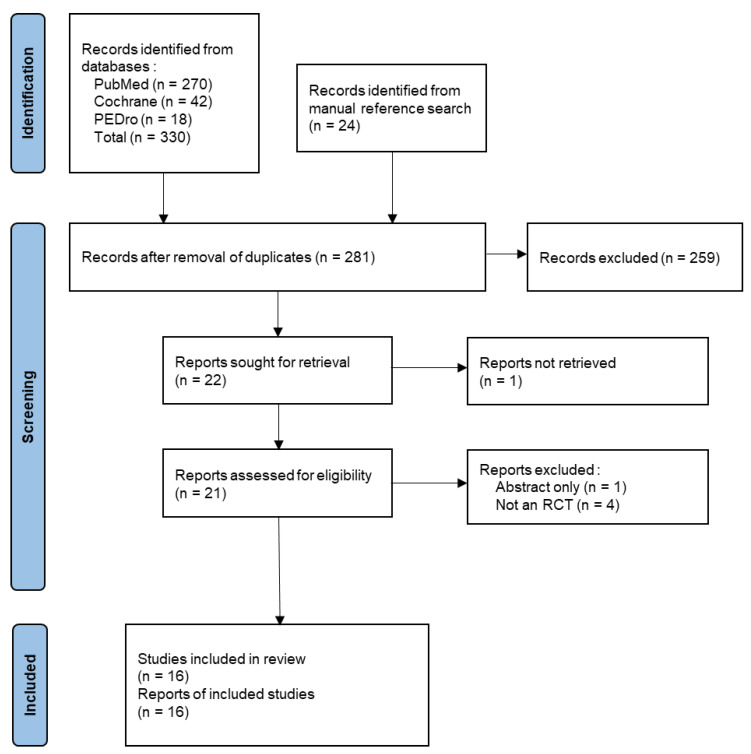
PRISMA Flowchart of the search and selection of the articles.

**Figure 2 ijerph-19-13734-f002:**
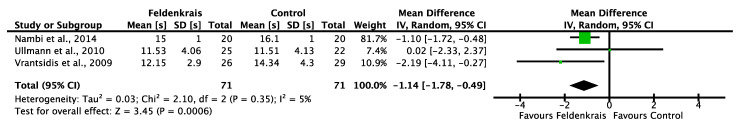
Meta-analysis of TUG in elderly subjects measured in seconds. (Nambi et al. [24], Ullmann et al. [20], Vrantsidis et al. [17]).

**Figure 3 ijerph-19-13734-f003:**
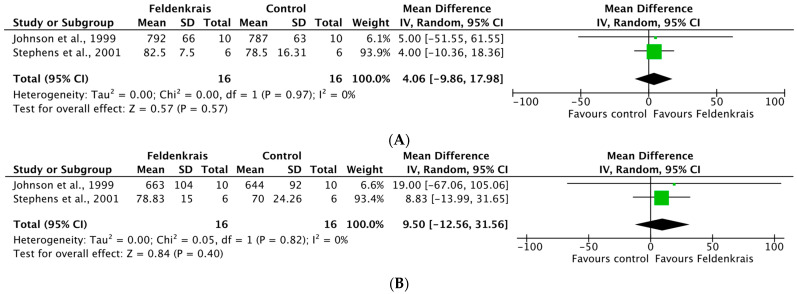
(**A**) Meta-analysis for the Function dimension of the MSSE scale in subjects with multiple sclerosis. (**B**) Meta-analysis for the Control dimension of the MSSE scale in subjects with multiple sclerosis. (Johnson et al. [22], Stephens et al. [26]).

**Table 1 ijerph-19-13734-t001:** Methodological quality of the included articles assessed by using the PEDro scale.

	Eligibility Criteria	Random Allocation	Concealed Allocation	Similar groups at Baseline	Subjects Blinding	Therapists Blinding	Assessors Blinding	Outcome Measures Obtained	Intention-to-Treat Analysis	Between-Group Statistical Comparisons	Point Measures and Measures of Variability	Total Score
Ahmadi et al., 2020 [13]	Yes	Yes	Yes	No	No	No	Yes	Yes	Yes	Yes	No	6/10
Chinn et al., 1994 [19]	No	Yes	No	No	No	No	No	Yes	Yes	No	Yes	4/10
Johnson et al., 1999 [22]	Yes	Yes	No	Yes	No	No	Yes	No	No	No	Yes	4/10
Lundblad et al., 1999 [23]	Yes	Yes	No	Yes	No	No	No	No	No	Yes	Yes	4/10
Lundqvist et al., 2014 [14]	Yes	Yes	No	Yes	No	No	Yes	No	Yes	Yes	Yes	6/10
Nambi et al., 2014 [24]	Yes	Yes	No	Yes	No	No	No	No	No	Yes	Yes	4/10
Palmer, 2017 [18]	Yes	Yes	No	No	No	No	Yes	No	No	Yes	No	3/10
Paolucci et al., 2017 [15]	No	Yes	Yes	Yes	No	No	Yes	Yes	No	Yes	Yes	7/10
Quintero et al., 2009 [25]	No	Yes	No	No	No	No	Yes	No	No	Yes	Yes	4/10
Smith et al., 2001 [16]	Yes	Yes	Yes	Yes	No	No	No	Yes	No	Yes	Yes	6/10
Stephens et al., 2001 [26]	Yes	Yes	No	Yes	No	No	No	No	No	Yes	Yes	4/10
Teixeira-Machado et al., 2015 [27]	No	Yes	No	Yes	No	No	No	Yes	No	Yes	Yes	5/10
Teixeira-Machado et al., 2017 [28]	No	Yes	No	Yes	No	No	No	Yes	No	Yes	Yes	5/10
Torres-Unda et al., 2017 [21]	Yes	Yes	Yes	Yes	No	No	No	No	No	Yes	Yes	5/10
Ullmann et al., 2010 [20]	Yes	Yes	No	Yes	No	No	No	No	Yes	Yes	Yes	5/10
Vrantsidis et al., 2009 [17]	Yes	Yes	Yes	Yes	No	No	Yes	Yes	No	Yes	Yes	7/10

**Table 2 ijerph-19-13734-t002:** Description of the included studies.

First Author (Year)	Study Objective	Type of Study	Population	Mean Age (SD)	Gender (M/F)	Groups (n)	Follow-Up and Outcomes Measures	Results
Quintero (2009) [25]	Evaluate the efficacy of FM to improve head posture and reduce nocturnal bruxism in children.	RCT with two parallel groups.	Children with bruxism.	4.73 (0.60)	NA	FG (13): ATM sessions, 3 h/once a week for 10 weeks. CG (13): Not specified.	Outcome measures: –Head posture (4 angles obtained from a lateral cephalometry) –Craniovertebral angle obtained from a photograph of the head Evaluation: –Pre-intervention –Post-intervention	There was a significant increase in the craniovertebral angle in the FG compared with the CG. The head posture in the FG after the intervention was less anterior and downward than the head posture in the CG.
Torres-Unda (2017) [21]	Evaluate the efficacy of FM to improve physical function and balance in middle-aged persons with intellectual disabilities.	RCT with two parallel groups.	Middle-aged subjects with intellectual disability	48.94 (6.01)	NA	FG (21, 16 analyzed): ATM sessions, 1 h/once a week for 30 weeks. CG (20, 16 analyzed): No intervention.	Outcome measures: –Physical Function Tests (SPPB) –Balance (stabilometry) Evaluation: –Pre-intervention –Just before the last session	SPPB: There was a significant improvement in the physical function of the FG compared with the CG (*p* < 0.01). Stabilometry: There was a significant decrease in the sway area in the FG (*p* < 0.05).
Lundqvist (2014) [14]	Evaluate the efficacy of FM to improve pain in persons with visual impairment and chronic neck/scapular pain.	RCT with two parallel groups.	Subjects with visual impairment and chronic neck/scapular pain.	53.3 (10.3)	10/51	FG (30): Combined sessions of ATM and FI, 2 h/once a week for 12 weeks. GC (31): No intervention.	Outcome measures: –Pain during palpation of the left and right occipital muscles, upper trapezius and levator scapulae muscles, measured with VAS. –Muscle complaints (subscale of the VMBC questionnaire) –Body pain (SF-36 subscale) Evaluation: –Pre-intervention –Post-intervention –One year after the intervention	Pain during palpation: There were significant between-group differences in the evolution of pain at post-intervention and at one-year follow-up. There were no significant changes in pain in the FG, while pain increased significantly in the CG. Muscular complaints: There were significant between-group differences in the evolution of the score at post-intervention. The score decreased significantly in the FG. Body pain: There were no significant differences.
Lundblad (1999) [23]	Investigate the effects of FM vs. physical therapy on neck and shoulder pain in industrial workers.	RCT with three parallel groups.	Women with neck or shoulder pain.	33 (9)	0/97	FG (33, 20 analyzed): Four sessions of FI and 12 sessions of ATM, 50 min/once a week, and home-based exercises for 16 weeks. PTG (32, 15 analyzed): Physical therapy sessions, 50 min/twice a week and home-based exercises for 16 weeks. CG (32, 23 analyzed): No intervention (waiting list).	Outcome measures: –Neck and shoulder ROM. –Estimated VO_2_ max during submaximal cycloergometry. –Endurance score: Sum of pain intensity (VAS) during a static shoulder flexion. –Cortical control score. –Physiological capacity based on isokinetic endurance test of the shoulder flexors on the dominant side (Surface EMG). –Measurement of painful neck and shoulder complaints: pain intensity (VAS), sick leave, prevalence and disability in leisure and work (questionnaires). Evaluation: –5 months before the intervention –1.5 months after the intervention.	In the FG, there were significant decreases in neck and shoulder complaints as well as in leisure disability. In the other two groups, there were either no changes (PTG) or complaints worsened (CG).
Chinn (1994) [19]	Evaluate the effect of one session of FM on functional reach of persons with neck, dorsal or shoulder pain.	RCT with two parallel groups.	Subjects with neck, dorsal or shoulder pain.	ND	1/22	FG (12): Follow the instructions of a 22-min audio of ATM related to neck and shoulders. A researcher made verbal and tactile clarifications if it was necessary. CG (11): Follow the instructions of a 16-min audio of simulated ATM related to neck and shoulders.	Outcome measures: –Functional Reach Test –Perceived exertion level during the Functional Reach Test measured with VAS. Evaluation: –Pre-intervention –Post-intervention	VAS: There was a significant reduction in the perceived exertion at post-intervention for the FG (*p* < 0.05). There were no significant differences in the CG. Functional reach: There were no significant differences in any group.
Paolucci (2017) [15]	Evaluate the efficacy of FM to reduce pain and improve interoceptive awareness in subjects with chronic low back pain.	RCT with two parallel groups.	Subjects with chronic low back pain.	FG: 61.21 (11.53) CG: 60.70 (11.72)	11/42	FG (26): ATM sessions, 1 h/twice a week for five weeks. CG (27): Physical therapy sessions (back school), 1 h/twice a week for five weeks.	Outcome measures: –Pain (VAS and MPQ) –Disability (WDI) –Quality of life (SF-36) –Mind-body interactions (MAIA) Evaluation: –Pre-intervention –Post-intervention –3 months after the start of the intervention	There were no significant between-group differences regarding the reduction of chronic pain. There was a correlation between the evolution of pain (VAS) and the Noticing subscale of the MAIA scale (R = 0.296, *p* = 0.037). There were significant changes in both groups in pain (*p* < 0.001) and disability (*p* < 0.001) over the investigation period.
Smith (2001) [16]	Determine the effect of one session of FM on pain and anxiety in people with chronic low back pain.	RCT with two parallel groups.	Subjects with chronic low back pain.	50.8 (16.2)	10/16	FG (14): Follow the instructions of a 30-min audio of ATM related to breathing. CG (12): Listen to a 30-min narration.	Outcome measures: –Pain (SF-MPQ) –Anxiety (STAI) Evaluation: –Pre-intervention –Post-intervention	Pain: There was a significant decrease in the affective dimension of pain in the FG. Anxiety: No significant differences.
Ahmadi (2020) [13]	Compare the effects of FM and core stability exercises on quality of life, pain, disability, interoceptive awareness and core musculature in subjects with chronic non-specific low back pain.	RCT with two parallel groups.	Subjects with chronic non-specific low back pain.	FG: 42.6 (11.6) CG: 38.89 (12.52)	NA	FG (30): ATM sessions, 30–45 min/ twice a week for five weeks. CG (30): Educational program and home-based core stability exercises with a prescribed progression for five weeks.	Outcome measures: –Quality of life (WHOQOL-BREF) –Pain (MPQ) –Disability (ODQ) –Interoceptive awareness (MAIA) –Diameter of the transversus abdominis muscle in contraction and at rest. Evaluation: –Pre-intervention –Post-intervention	There were statistically significant between-group differences for quality of life (*p* = 0.006), interoceptive awareness (*p* < 0.001) and disability (*p* = 0.021) in favor of the FG. Pain: McGill’s pain score decreased significantly in both groups, but there were no significant between-group differences. Transversus abdominis diameter at rest and in contraction: There was a significant increase in both groups, but the increase was significantly greater in the CG.
Johnson (1999) [22]	Evaluate the efficacy of FM to improve the emotional status and function in subjects with multiple sclerosis.	RCT with crossover design.	Subjects with multiple sclerosis.	44.8 (1.4)	5/15	FG (10): ATM Sessions, 45 min/once a week for eight weeks. CG (10): Simulated manual treatment, 1 h/ once a week, for eight weeks.	Outcome measures: –Manual dexterity test (9HPT) –Anxiety and depression (HAD) –Confidence in functional ability (MS Self-Efficacy Scale) –Symptoms scale (MS Symptom Inventory) –Functionality scale (MS Performance Scales) –Scale of perceived stress Evaluation: –Pre-intervention –Between interventions –Post-intervention	Significant differences in perceived stress, as well as a tendency to reduce anxiety, were reported after the Feldenkrais sessions. MS Self-Efficacy Scale: There were non-significant trends towards greater self-efficacy after both Feldenkrais and simulation sessions. There were no differences in the other measures.
Stephens (2001) [26]	Determine the efficacy of FM to improve balance, balance confidence and functional capacity confidence in persons with multiple sclerosis.	RCT with two parallel groups.	Subjects with multiple sclerosis.	FG: 56.2 (9.9) CG: 51.8 (10.2)	4/8	FG (6): 8 ATM sessions, 2–4 h (20 h in total) over a ten weeks period. CG (6): Four 90-min educational sessions by experts in multiple sclerosis over a ten weeks period.	Outcome measures: –Balance (fall register, Equiscale, mCTSIB and LOS) –Balance confidence (ABC) –Functional capacity confidence (MS Self-Efficacy Scale) Evaluation: –Pre-intervention –Post-intervention	There was a statistically significant increase in the mCTSIB score in the FG; the FG had significantly fewer abnormal mCTSIB tests and demonstrated better balance confidence compared with the CG. There was a trend towards improvement in all other measures in the FG compared with the CG.
Teixeira-Machado (2015) [27]	Determine the efficacy of FM to improve quality of life and depression in elderly patients with Parkinson’s disease.	RCT with two parallel groups.	Elderly patients with Parkinson’s disease.	FG: 60.70 (2.55) CG: 61 (2.70)	NA	FG (15): 50 ATM sessions, 1 h/twice a week. CG (15): Educational reading about fall prevention, medication and daily life management.	Outcome measures: –Quality of life (PDQL) –Depression (BDI) –Cognitive status (MMSE) Evaluation: –Pre-intervention –Post-intervention	There was a significant improvement in quality of life scores (*p* = 0.004) as well as a reduction in the level of depression (*p* = 0.05) in the FG compared with the CG. The mental state score increased significantly in the FG (*p* < 0.001) and decreased in the CG (*p* = 0.04).
Teixeira-Machado (2017) [28]	Evaluate the efficacy of the exercise based on FM to change the functional capacity of elderly patients with Parkinson’s disease.	RCT with two parallel groups.	Elderly patients with Parkinson’s disease.	GF: 60.70 (2.55) GC: 61 (2.70)	NA	FG (15): 50 ATM sessions, 1 h/twice a week. CG (15): Educational reading about fall prevention, medication and physical activity.	Outcome measures: –Functional tests to assess balance, mobility, strength and gait speed (walk in a figure-eight trajectory, TUG, lying rollover, standing 360° turn-in-place, functional reach, sitting/standing, BBS and hip flexion strength). Evaluation: –Pre-intervention –Post-intervention	There were significant differences between groups in the evolution of the functional test score. In all tests, the FG performed significantly better (*p* ≤ 0.05) compared with the CG.
Vrantsidis (2009) [17]	Evaluate the effects of an ATM program to improve balance and function in elderly patients.	RCT with two parallel groups.	Elderly patients.	74.9 (8.2)	13/42	FG (29, 26 analyzed): ATM sessions, 40–50 min/ once a week for eight weeks. CG (33, 29 analyzed): No intervention (waiting list).	Outcome measures: –Activities questionnaire (Frenchay Activity Index and Human Activity Profile) –Quality of life (AQoL) –Fear of falling (Modified FES) –Cognitive status (Abbreviated Mental Test Score) –Functional tests to evaluate balance, gait and function (FSST, TUG, Step Test, Timed Sit-to-Stand Test, gait speed and duration of the double-support phase). –Force platform measures assessing gait, balance and function. Evaluation: –Within three weeks before the intervention. –Within two to three weeks after the intervention.	There was a significant improvement for the FG compared with the CG in the Modified FES score (*p* = 0.003) and gait velocity (*p* = 0.028), as well as a strong tendency for improvement in the TUG score (*p* = 0.056). There were no significant between-group differences in the other measures.
Ullmann (2010) [20]	Determine the efficacy of FM to improve balance, mobility, gait and balance confidence in elderly patients.	RCT with two parallel groups.	Elderly patients.	75.6 (7.3)	14/33	FG (25): ATM sessions, 1 h/3 times per week for five weeks. CG (22): No intervention (waiting list).	Outcome measures: –Balance (Tandem test) –Mobility (TUG and TUG with cognitive tasks) –Gait characteristics (GAITRite Walkway System) –Balance confidence (ABC) –Fear of falling (FES) Evaluation: –Pre-intervention –Post-intervention	There was a significant improvement for the FG compared with the CG in balance (*p* = 0.03), mobility (*p* = 0.042) and fear of falling (*p* = 0.042). No other significant changes were reported. However, FG participants showed improvements in balance confidence (*p* = 0.054) and TUG with added cognitive task (*p* = 0.067).
Nambi (2014) [24]	Compare the efficacy of Pilates Method and FM to improve functional balance, mobility and quality of life in elderly persons.	RCT with three parallel groups.	Elderly patients.	G: 70.4 (2.8) PIG: 70.8 (2.8) CG: 69.35 (3.0)	37/23	FG (20): ATM sessions, three times a week for six weeks. GPI (20): Pilates exercises three times a week for six weeks. CG (20): A program consisting of 5 min of warm-up, 12 min of walking at a comfortable speed and 5 min of cool-down. Three times a week for six weeks.	Outcome measures: –Forward reach test –Mobility (TUG) –Functional Balance (Dynamic gait index) –Quality of life (RAND-36) Evaluation: –Pre-intervention –Post-intervention	In the FG and GPI, there was a significant improvement in all measures (*p* ≤ 0.001). However, GPI scored clinically better compared with the FG in all measures. In the CG there were significant improvements in the TUG (*p* = 0.022) and Dynamic Gait Index (*p* = 0.042) scores.
Palmer (2017) [18]	Evaluate the FM efficacy to improve balance, mobility and functional capacity in elderly patients.	RCT with two parallel groups.	Elderly patients.	76	16/108	FG (70, 45 analyzed): ATM sessions, either 2 h/twice a week for six weeks or 2 h/once a week for twelve weeks. CG (54, 36 analyzed): No intervention (waiting list).	Outcome measures: –Forward reach test –Mobility (TUG) –Balance (Base of support and tandem posture) –Difficulty in performing tasks (OPTIMAL modified) –Self-determined questionnaire on individual priorities and the effectiveness of the intervention Evaluation: –Pre-intervention –Post-intervention	There were significant correlations between the number of attended lessons and both functional reach test and modified OPTIMAL scores. A significantly higher proportion of the FG (versus CG) reported positive changes in the self-determined questionnaire in both prioritized and newly identified activities.

M/F, Male/Female; RCT, Randomized Controlled Trial; FM, Feldenkrais Method; ATM, Awareness through movement; FI, Functional Integration; FG, Feldenkrais group; CG, Control Group; PTG, Physical Therapy Intervention Group; PIG, Pilates Intervention Group; NA, not available; SPPB, Short Physical Performance Battery; VAS, Visual Analogue Scale; VMBC, Visual, Musculoskeletal, and Balance Complaints; SF-36, 36-Item Short-Form Health Survey; ROM, Range of movement; VO2max, Maximum oxygen volume; EMG, Electromyography; MPQ, McGill Pain Questionnaire; WDI, Waddel Disability Index; MAIA, Multidimensional Assessment of Interoceptive Awareness Questionnaire; SF-MPQ, Short-Form McGill Pain Questionnaire; STAI, State Trait Anxiety Inventory; WHOQOL-BREF, World Health Organization’s quality of life instrument short form; ODQ, Oswestry Disability Questionnaire; 9HPG Nine-Hole Peg Test; HAD Hospital Anxiety and Depression Scale; MS Multiple sclerosis; mCTSIB, Basic Balance Master modified Clinical Test of Sensory Interaction in Balance; LOS Limits of Stability; ABC, Activities-specific Balance Confidence Scale; PDQL, Parkinson’s Disease Quality of Life Questionnaire; BDI, Beck Depression Inventory; MMSE, Mini Mental State Questionnaire; TUG, Timed-Up-and-Go Test; BBS, Berg Balance Scale; AQoL, Assessment of Quality of Life instrument; FES, Falls Efficacy Scale; FSST, Four Square Step Test; RAND-36, RAND 36-Item Short Form Survey Instrument; OPTIMAL, Outpatient Physical Therapy Improvement in Movement Assessment Log.

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
