# Peer review of "Effects of the Feldenkrais Method as a Physiotherapy Tool: A Systematic Review and Meta-Analysis of Randomized Controlled Trials"

_ijerph, 2022, doi:10.3390/ijerph192113734_

Round 1

Reviewer 1 Report

Contributions of the paper should be highlighted.

The authors should clearly state the parameters of the study.

Figures should be cited close to its reference.

Resolution of the images should be improved.

Author Response

The authors would like to thank the editor and the reviewers for their effort to carefully review our manuscript, as well as for providing us with their comments and suggestions to improve the quality of the same. According to their suggestions, we believe the paper has been improved and clarified: the following responses have been prepared to address the reviewers’ comments in a point-by-point fashion.

Comment 1#: Contributions of the paper should be highlighted.

Response: We thank the reviewer for the comment, but we would like to explain that the contributions of the authors have been explained in the appropriate section at the end of the manuscript (“Authors contributions”). Moreover, the first word of the name/last name of those authors who participated in the peer review process, as well as the risk of bias assessment have been written in the Methods section.

Comment 2#: The authors should clearly state the parameters of the study.

Response: We thank the reviewer for the suggestion. Firstly, the authors carried out a previous rapid review, and found that the number of articles of the Feldenkrais method was limited. Therefore, the authors decided to carry out a systematic review including all those articles that measured any outcome related to physical condition. The authors established several criteria to determine which outcomes were considered as physical condition. In this regard, we agree with the reviewer that this was not completely clear in the manuscript, and therefore these criteria have been added to the manuscript in the "Eligibility criteria" section.

Comment 3#: Figures should be cited close to its reference.

Response: Thank you for the comment. We have inserted the figures close to its references, in order to facilitate understanding.

Comment 4#: Resolution of the images should be improved.

Response: Thank you for the suggestion. We have reprocessed the figures to increase their quality. If needed, we could also send high quality images to the journal as separated images.

Reviewer 2 Report

The article presents a well-developed, complete, and adequately supported introduction. The objectives of the study and purposes of the manuscript are stated.

The materials and methods are described in detail, providing not only an excellent tool that guarantees the reproducibility of the study, but also an excellent example of a well-designed systematic review.

The statistical methods are adequate, and the tables and figures help to describe the methodology and studies considered. Table 1 and 2 are particularly clear, synthetic and at the same time complete, guaranteeing the quality of the chosen data.

Figure 1 conforms to international standards to explain the processes developed.

The results are well exposed, with numerical data and clear and well-chosen statistics. There are no subjective comments. All the data presented in the section are objective and substantiated, without important biases.

The discussion is relatively brief, though appropriate. Describe the limitations well.

It would be appropriate to also describe the strengths: first work of its kind on the subject, statistical rigor, etc.

The abstract should contain some numerical data on the main results. Except for this small detail, the work could be published without problems in my opinion.

Author Response

The authors would like to thank the editor and the reviewers for their effort to carefully review our manuscript, as well as for providing us with their comments and suggestions to improve the quality of the same. According to their suggestions, we believe the paper has been improved and clarified: the following responses have been prepared to address the reviewers’ comments in a point-by-point fashion.

Comment 1#: The article presents a well-developed, complete, and adequately supported introduction. The objectives of the study and purposes of the manuscript are stated. The materials and methods are described in detail, providing not only an excellent tool that guarantees the reproducibility of the study, but also an excellent example of a well-designed systematic review. The statistical methods are adequate, and the tables and figures help to describe the methodology and studies considered. Table 1 and 2 are particularly clear, synthetic and at the same time complete, guaranteeing the quality of the chosen data. Figure 1 conforms to international standards to explain the processes developed. The results are well exposed, with numerical data and clear and well-chosen statistics. There are no subjective comments. All the data presented in the section are objective and substantiated, without important biases. The discussion is relatively brief, though appropriate. Describe the limitations well. It would be appropriate to also describe the strengths: first work of its kind on the subject, statistical rigor, etc.

Response: We sincerely appreciate the positive feedback received. We have added the strengths of the study.

Comment 2#: The abstract should contain some numerical data on the main results. Except for this small detail, the work could be published without problems in my opinion.

Response: Thank you for the comment. We have added some numerical data in the results of the abstract.